# *Escherichia coli* Extract-Based Cell-Free Expression System as an Alternative for Difficult-to-Obtain Protein Biosynthesis

**DOI:** 10.3390/ijms21030928

**Published:** 2020-01-31

**Authors:** Sviatlana Smolskaya, Yulia A. Logashina, Yaroslav A. Andreev

**Affiliations:** 1Institute of Molecular Medicine, Sechenov First Moscow State Medical University, Trubetskaya str. 8, bld. 2, 119991 Moscow, Russia; yulia.logashina@gmail.com (Y.A.L.); yaroslav.andreev@yahoo.com (Y.A.A.); 2Shemyakin-Ovchinnikov Institute of Bioorganic Chemistry, Russian Academy of Sciences, ul. Miklukho-Maklaya 16/10, 117997 Moscow, Russia

**Keywords:** recombinant expression, cell-free (CF) protein synthesis, membrane proteins, disulfide bond, protein folding, noncanonical amino acids (NAAs), post-translational modification (PTM)

## Abstract

Before utilization in biomedical diagnosis, therapeutic treatment, and biotechnology, the diverse variety of peptides and proteins must be preliminarily purified and thoroughly characterized. The recombinant DNA technology and heterologous protein expression have helped simplify the isolation of targeted polypeptides at high purity and their structure-function examinations. Recombinant protein expression in *Escherichia coli*, the most-established heterologous host organism, has been widely used to produce proteins of commercial and fundamental research interests. Nonetheless, many peptides/proteins are still difficult to express due to their ability to slow down cell growth or disrupt cellular metabolism. Besides, special modifications are often required for proper folding and activity of targeted proteins. The cell-free (CF) or in vitro recombinant protein synthesis system enables the production of such difficult-to-obtain molecules since it is possible to adjust reaction medium and there is no need to support cellular metabolism and viability. Here, we describe *E. coli*-based CF systems, the optimization steps done toward the development of highly productive and cost-effective CF methodology, and the modification of an in vitro approach required for difficult-to-obtain protein production.

## 1. Introduction

*Escherichia coli* is considered as a first-choice heterologous host organism for the expression of recombinant protein. The advantages of *E. coli* as an organism for recombinant protein expression and purification are well known and rely on simplicity, fast growth, and cost-effective procedures of cell cultivation and protein extraction. Despite the availability of diverse *E. coli* strains, a variety of expression plasmids, and the number of approaches developed to produce different recombinant peptides and proteins [1,2], the application of such a heterologous host is still limited or not even possible for a broad range of peptides and proteins. The examples of difficult-to-obtain proteins include eukaryotic ones, whose synthesis requires codon optimization or post-translational modification, disulfide-bonded, toxic and integral membrane peptides and proteins. Moreover, screening for optimal strategy for the desired protein synthesis can be time-consuming, laborious, and expensive.

Since it has been identified that protein synthesis is a ribosome-mediated process [3] that does not require the integrity of the cell [4], cell-free (CF) or in vitro protein synthesis is considered a good alternative for recombinant protein production. The productivity of cell-free expression systems has been greatly improved and in vitro methodology has been used for a variety of applications. The CF protein synthesis systems are generally constructed with cell extract prepared from *E. coli*, wheat germ, insect cells, or rabbit reticulocytes. The *E. coli* extract-based CF expression system is the most popular, and broad varieties of in vitro protein synthesis systems are commercialized by various companies (e.g., Biotechrabbit, Invitrogen, Qiagen, Promega), and have been used for a range of applications.

The CF protein synthesis system has several advantages over current in vivo processes. Firstly, since there is no need to support cellular metabolism; all of the cellular resources can be efficiently directed toward the production of a single protein [5]. Although the coupled (i.e., combining both transcription and translation processes) CF system has been proven to be more efficient than the uncoupled one [6], it is still possible to use mRNA or PCR fragments as the matrices escaping genetic engineering and cloning procedures [7]. Taken together, this makes CF technology a reliable and fast way to obtain a high yield of the desired protein. Secondly, the reaction environment can be directly controlled and easily manipulated, since there is no cell barrier. This open nature of the CF system promotes the synthesis of peptides and proteins, whose intracellular expression is problematic, by supplying a CF reaction mixture with the required additives. These additives include chaperones and reducing agents, stabilizing mRNA or promoting complex protein folding, hydrophobic compounds required for soluble expression of membrane proteins (MPs), rare tRNA molecular species for codon usage bias compensation, and orthogonal aminoacyl-tRNA synthetase (aaRS)/tRNA pairs (orthogonal translation system, OTS) for site-specific non-canonical amino acids (NAAs) incorporation in response to unique codon (Figure 1). These additives are often not applicable for in vivo protein expression due to their inability to cross the cellular membrane or due to their toxic effect on cellular metabolism. Finally, the overexpression of some proteins, including integral membrane proteins (MPs) or incorrectly folded proteins forming aggregates, negatively effects cell fitness and viability. There is also a group of toxic genes that destroy cellular membrane integrity or disrupt the synthesis of nucleic acids, proteins, or cell wall of bacterial host organisms. In this instance, the in vitro approach is a salvation for the synthesis of moderately or highly toxic peptides or proteins.

## 2. History

The development of CF synthesis technology has been directed toward the manufacturing of reliable and efficient methodology allowing for high protein yield production. The establishing of *E. coli* cell’s genetics and metabolism, determining growth parameters, and inventing procedures for cell extract preparation has led to the enhanced efficacy of in vitro methodology.

The CF protein synthesis system was first carried out using the extracts of rat liver cells during the investigation of the in vitro incorporation of radiolabeled amino acids into proteins in the 1950s [8,9]. Due to success in the preparation of different extract from mammalian cells, scientists managed to identify ATP and GTP requirements for protein production, specify ribosomes as the synthesis stations, and isolate tRNA and amino acid-loading enzymes [3,10,11,12]. Similar CF protein synthesis systems from *E. coli* cells were prepared only in 1960 due to difficulties in bacterial cell wall destruction [13]. Additionally, the ability to synthesize a protein in an *E. coli* extract-based reaction mixture strongly relies on the presence of sufficient amounts of amino acids, potassium and magnesium ions, energy sources (ATP, GTP), along with a system of energy regeneration. Subsequently, the first CF synthesis system for polyphenylalanine production based on *E. coli* extract was developed in 1961. As a result, the first genetic codon UUU encoding its cognate amino acid phenylalanine were identified [14]. This breakthrough gave rise to the deciphering of the genetic code [15] and establishing the “central dogma of molecular biology” [16,17], and also made CF protein synthesis an important tool that has been widely applied in structural and molecular biology.

The next step in the development of CF protein synthesis system was the expression of foreign genes in *E. coli* extracts using DNA as the template, instead of mRNA [18,19]. The coupled transcription-translation system was first used in studies on operon (*lac*, *ara*, *gal*, *trp*, and *arg*) regulation. The RNA polymerase has been proven to be an essential enzyme in the protein synthesis process catalyzing DNA transcription into mRNA. Thereafter, cell extracts have been customized for coupled transcription-translation synthesis by genetic engineering approaches. The usage of more efficient phage promoters and RNA polymerases (T7 or SP6) increased the production of mRNA and, consequently, magnified the obtained protein yield [20,21]. Further enhancement of CF system productivity was achieved by stabilization of RNA and linear DNA by removing genes encoding RNase I or E, and exonucleases V and I [22,23,24]. Interestingly, an extract prepared from *E. coli* grown at low temperature has been demonstrated to provide higher protein productivity because of the suppressed exonuclease V expression [25]. Moreover, insertion of χ-site into linear DNA [26] or T7 terminator at the 3′-end of the mRNA [22] enhanced the productivity of CF synthesis due to DNA or RNA stabilization, correspondingly. Another approach to improve linear DNA stability without genomic engineering of host strain is based on PCR-product cyclization by self-ligation [27].

Methods for cell extract preparation have also changed over time. The standard protocol for making the *E. coli* extract was developed in 1984 [28]. It involved complicated steps of cell culture fermentation, French press cell lysis, high-speed centrifugation, 80 min run-off reaction, 180 min dialysis, and final low-speed centrifugation. The first essential modification that simplified the technique was made in 2006 and resulted in a decrease by two-times of the required steps of extract production [29,30]. Since the crude *E. coli* extract has been demonstrated to be applicable for CF protein synthesis, the improved protocol consisted of fermentation, French press cell lysis, low-speed centrifugation, and 30 min run-off reaction. Furthermore, while the laboratory equipment was upgraded (flask shaker and sonicator used instead of the fermenter and French press) and other techniques of cell disruptions became common (like chemical and freezing/thawing), cell extract preparation turned to be an accessible and efficient procedure [31,32,33].

Although the efficiency and productivity of the extract-based CF technology have been greatly improved over the last two decades, the development of the reconstituted minimal translational machinery of *E. coli*, so-called PURE (protein synthesis using recombinant elements), has revolutionized the CF protein synthesis field [34,35]. This system is composed of purified His-tagged version of all protein factors involved in the initiation, elongation, termination (IET, encoded by 11 genes) and aminoacylation processes (aaRSs, encoded by 23 genes), and 46 essential tRNA molecular species [35]. PURE is a highly purified protein synthesis system that enables to overcome problems arising from the usage of cell extracts, i.e., nucleases and proteases that degrade substrates and products, respectively, and low controllability of transcription/translation processes due to the presence of unknown or poorly characterized components of cell extracts. The PURE system is commercially available; as well as its withdrawn variants lack one or several translation factors (Figure 2). Nonetheless, the application of the PURE system is limited: the necessity to clone, express, and purify 34 individual His-tagged proteins restricts routine production of the PURE system in the laboratory and makes commercial analogous highly expensive. In attempts to simplify the process of PURE system component preparation, poly-histidine tags were introduced in gene encoding for the translation machinery of *E. coli* by the multiplex automated genome engineering (MAGE) approach using short oligonucleotides [36]. This approach allowed the efficient incorporation of up to eight His-tags without deleterious effects on cell viability and simultaneous co-purification of functional proteins. The recent progress in PURE system component purification was achieved by the simultaneous expression and co-purification of all the 34 His-tagged proteins in the engineered synthetic microbial consortia cultivated in a single bacterial culture [37]. The consortia consisted of the individual *E. coli* strains expressing up to 3 His-tagged translation machinery proteins. The precise ratio of the PURE system components was tightly controlled by initial bacterial density, plasmid copy numbers, and transcription-translation rate regulation. Taken together, the synthetic microbial consortia application made the PURE system production relatively simple, fast, and cost-effective.

### 2.1. Energy Regeneration

Efficient and successful CF protein synthesis is dependent on a sufficient amount of consumable substrates, such as amino acids and ATP. The incensement of ATP quantities is a challenge addressed to ATP regeneration systems [38]. Classical ADP phosphorylation is achieved using substrates with high-energy phosphate bonds, like phosphoenolpyruvate, acetyl phosphate, or creatine phosphate along with corresponding kinases (such as pyruvate kinase, acetate kinase, or creatine kinase). The degradation of high-energy substrates in the CF system induced the accumulation of inorganic phosphate, followed by chelation of magnesium ions, turning an important translational co-factor into magnesium phosphate [39]. For further improvement of energy supply, researchers used pyruvate, which first was converted into acetyl phosphate by pyruvate oxidase utilizing inorganic phosphate, and then acetyl phosphate was applied as the high-energy source for ATP regeneration accompanied by equimolar inorganic phosphate release [40]. Later, scientists realized that carbon metabolism could be reconstituted in the CF system, and glucose oxidation produced sufficient quantities of energy essential for ADP phosphorylation [41,42]. However, glucose oxidation is accompanied by rapid protons accumulation and, consequently, pH decrease that inhibits protein synthesis. To overcomes this issue, three strategies were developed: first, the application of Bis-Tris buffer could maintain the reaction pH [41]; second, the usage of slow secondary sources of glucose such as maltodextrin and starch could provide long-term protein synthesis [43,44,45]; and third, the supply of pH-activated glutamate decarboxylase converting glutamate into γ-aminobutyric acid utilize protons [46].

### 2.2. Formats of CF Systems

The productivity of the CF system has been limited not only by the consumption of ATP but also by the utilization of other substrates required for protein biosynthesis reactions, such as NTPs and amino acids. The most commonly used CF synthesis is conducted in the batch format (Figure 3a), utilizing one reaction vessel for the transcription/translation processes. The productivity of the batch format can be augmented by continuous supply of the reaction medium to the amino acids, thus providing prolonged translation. On the other hand, in vitro protein synthesis can be inhibited due to the accumulation of by-products in the reaction medium. To further enhance the yield of in vitro synthesized protein, the dialysis CF format, which is based on the separation of the reaction mixture from the feeding solution comprising all the low-molecular-weight substrates, was invented [47]. The continuous exchange between the reaction mixture and the feeding solution providing the substrates supply and synthesized protein or by-product removal happens either actively by a flow (so-called continuous-flow CF systems) or passively by dialysis through the semi-permeable membrane (continuous-exchange CF system) (Figure 3b,c) [48]. The Endo laboratory developed a bilayer diffusion system, simplified continuous-exchange methodology, by modifying the batch format [49]. The bilayer CF system is composed of a reaction mixture covered with a feeding solution that allows protein synthesis in the interphase and enables substrate supply and by-product removal. In general, the application of the dialysis CF format extends the polypeptide chain synthesis reaction and yields about mg per mL of the desired protein (Figure 3d).

Another way to continuously supply the CF reaction medium with substrates and energy sources as well as to remove reaction by-products is based on the application of microfluidic technology [50]. The usage of microfluidic devices helps save resources required for in vitro protein synthesis, provide accurate mixing of reaction components, and maintain optimal conditions. The microfluidic droplet technology has been widely used to create artificial cells by encapsulating (compartmentalizing) cell lysate for reaction mixture separation from the feeding solution. The physical border can consist of a surfactant monolayer [51] or lipid mono- or bilayer [52,53]. The application of microfluidic devices and artificial cells allows to enhance the productivity of the CF system and provides powerful tools for fundamental research, synthetic biology, and biomedical usage.

## 3. Making Difficult-to-Obtain Protein Synthesis Easy

### 3.1. Cytotoxic Protein and Peptides Therapeutics

Unfortunately, many therapeutic proteins have pronounced toxicity because of their natural activity. For instance, many cancer therapeutics are toxic to eukaryotic cells and cannot be expressed in heterologous prokaryotic cells due to codon biases or improper folding and protein aggregation. An example of effective in vitro system application is the production of onconase, ribonuclease with predominant activity toward tRNAs, in an *E. coli* CF reaction mixture supplied with externally prepared tRNA molecules [54]. Another group of the therapeutics of interest, which is considered as a promising alternative to the traditional antibiotics, is a group of antimicrobial peptides. The effective synthesis of active recombinant antimicrobial peptides in the correct spatial form is possible only in vitro, since the diverse range of these therapeutics activities [55] leads to bacterial cell growth attenuation or death. Successful application of CF technology was demonstrated in the production of peptide therapeutics, such as cecropin [56] and beta-defensin-2 [57], as well as for fast screening and selection of active antimicrobial peptide from random peptide library [58].

### 3.2. Membrane Protein Synthesis

Integral MPs play an essential role in living cells; they participate in signal transduction, cellular and energetic metabolism, transport processes, and many other vital functions, making them one of the most promising targets of modern drug delivery [59]. However, the properties of MPs make them one of the most difficult proteins to produce using cellular expression, which, in turn, prohibits proper investigation of their structure and function. The synthesis of recombinant MPs in diverse host organisms is often accompanied by incorrect folding and aggregation due to the hydrophobic nature of MPs. Proper integration of recombinant MPs requires correct targeting and translocation into membranes, which is not always possible in heterologous host cells. The overexpression of MPs has a toxic effect due to the disintegration of host cell membranes or disruption of cellular signaling or transport function [60]. In this context, the open nature of CF systems allows to escape cytotoxicity and manipulate the reaction medium, thus solving most of the MPs synthesis problem. The synthesis of MPs as a precipitate or as soluble proteins in CF systems can be performed in the absence or presence of detergents/lipids, respectively (Figure 4).

In the absence of any hydrophobic supplement in the CF reaction medium, the hydrophobic domains of the newly synthesized MPs interact with each other forming insoluble aggregates and precipitate in the reaction medium (P-CF) (Figure 4a) [61]. Although MPs produced by the P-CF methodology are not correctly folded and inactive, their spatial structure and activity can be reconstituted in the presence of appropriate detergents/lipids, allowing for protein solubilization and micelles/liposomes formation. In contrast to the inclusion bodies formed after in vivo expression, the precipitates of MPs formed in P-CF can be solubilized quickly and efficiently in mild detergents due to relatively weak protein-protein interaction [62]. It is also possible to reconstitute precipitated MPs in lipids by freezing-thawing in the presence of liposomes made from phosphatidylserine, phosphatidylcholine, and cholesterol among others [63,64]. The *E.coli*-based P-CF was successfully used to produce a number of MPs, such as G protein-coupled receptors, GPCRs, [65], *E. coli* small multidrug transporters comprising from 4 to 6 transmembrane domain [64,66], *Bacillus subtilis* phospho-*N*-acetyl-muramoyl-pentapeptide translocase, MraY, involved in peptidoglycan layer biosynthesis [67], and many others [68], whose synthesis in cell-based expression systems was difficult or even not possible. Clearly, the P-CF cannot be considered a general technology for MPs’ protein production, since selecting the optimal detergent or lipid composition for MPs solubilization can be an extended and laborious process that does not always provide correct folding and full reconstitution of catalytic activity [69,70]. Still, P-CF can be used as a fast and reliable screening approach to test the possibility to express MPs.

The possibility to directly modify the reaction mixture of CF expression system by supplying it with the desired hydrophobic compounds is widely explored to produce MPs in a soluble form. A number of detergents [70,71,72] can be used to generate a hydrophobic environment of CF (D-CF) and facilitate the insertion of synthesized protein into the micelles (Figure 4b). The choice of an appropriate amphiphilic agent depends on the individual properties of the MP of interest. In general, the D-CF technology usually utilizes mild detergents with low critical micelle concentration, CMC, a specific threshold concentration of detergent required for the micelle’s formation. Hence, such detergents should not interfere or disrupt transcription-translation processes, prevent MPs from aggregation, and have no effect on the catalytic activity of MPs [73]. The classical detergents with an extremely low CMC, such as long-chain derivatives of polyoxyethylene-alkyl-ethers, Brij, variants from Tween-series, n-dodecyl-β-d-maltoside, DDM, Digitonin, and Triton X-100 have been successfully used for MPs synthesis in the D-CF mode [70,74,75,76]. Since there is no need to solubilize MPs from the membrane, many other hydrophobic compounds with no lipophilic properties can be supplied to D-CF for soluble MPs synthesis. Among them are synthetic fluorinated and hemifluorinated surfactants [77,78,79], nonionic amphipols with high molecular mass [78,80], and lipid-like peptides [81,82]. Overall, the D-CF approach is a simple and convenient way to co-translationally solubilize MPs. The disadvantages of D-CF methods are often low yield of MPs compared to that in precipitates, the necessity to select an appropriate detergent that allows correct folding and activity of soluble MPs, and the need to transfer MPs from micelles to lipid bilayers for further functional and structural investigations [83].

The maintenance of spatial form, activity, and stability of many MPs requires insertion into the lipid bilayer or depends on the presence of a particular lipid in the membrane. The CF synthesis in lipidic environments (L-CF) represents more reliable and accurate technology, allowing for proper MPs folding and functionality (Figure 4c). One of the L-CF approaches relies on the direct insertion of synthesized MPs into bicelles attained by supplying the reaction medium with a mixture of lipids and one or more detergents [61,84]. The MPs produced in vitro are inserted into mixed detergent-lipid micelles, but since the detergent’s molecules can be easily removed from the reaction mixture to external solutions by dialysis, the mixed micelles turn into bicelles (lipid bilayer discs surrounded by detergents) and liposomes (lipid bilayer vesicles) as the concentration of detergents in the reaction mixture get lesser. The detergent-lipid mode of L-CF synthesis promotes correct folding of MPs that has been demonstrated for membrane subunits a and c of the *E. coli* ATP-synthase, which had a similar structure as those ones isolated from the bacterial membrane [85]. However, this mode of L-CF is characterized by overall low yield of MPs [74] and complexity resulting from the necessity to select both suitable detergents and lipid composition [61].

Another L-CF approach was successfully applied for MPs synthesis in the presence of liposomes, biological membranes, and nanodiscs (nanolipoproteins particles, NLP) with no detergent used. The insertion of MPs into the lipid bilayer can occur directly by an undefined mechanism; the efficacy of MPs insertion depends greatly on the properties of lipids composing the membrane and phase transition temperature [86,87,88]. Another way of protein insertion into the membrane relies on the natural translocation mechanism. Majority of *E. coli* MPs proteins inserted into the membrane by the Sec-dependent pathway are determined by the signal recognition particle (SRP) and its cognate receptor (SR), and SecYEG-SecDFYajC-YidC insertion apparatus [89]. The insertion of MPs by L-CF exploring the natural translocon machinery can be achieved by supplying the reaction medium with cytosolic SecA and SRP factors [90] and microsome fraction prepared from the *E. coli* membrane [91], comprising of integrated SR, SecYEG translocon, and YidC insertase, or by direct in vitro synthesis of SecYEG translocon by expressing encoding cognate genes [92]. Application of liposomes and microsomes for the structural study of MPs is limited by the relative heterogeneity of lipid vesicles. In this context, the application of NLP, representing a lipid bilayer of discoidal shape wrapped by two copies of apolipoprotein or membrane scaffold protein [93], has been gaining attention. The size of NLPs is precise (ranging from 9 to 20 nm) and determined by the type of scaffold protein [94]. Another advantage of NLPs application for MPs production arises from the possibility to tag scaffold proteins, simplifying the purification procedure. The L-CF synthesis of MPs in the presence of nanodiscs can be achieved by either supplying the reaction medium with preformed NLPs [95,96] or by simultaneous in vitro expression of both MP and scaffold protein [97].

In summary, despite difficulties arising from the necessity to screen appropriate hydrophobic compounds and isolate and prepare membrane vesicles or NLPs, in vitro methodology has been successfully applied in the synthesis and characterization of such difficult-to-obtain MPs as bacterial membrane transporters [64,66,67], G-PCR [65,98], and photosystem II subunit S, PsbS [99], among other uses. CF technology is considered by the protein researcher community as a perspective technique to produce MPs in a stable form, allowing for further functional and structural investigation.

### 3.3. CF Synthesis of Folded Proteins and Peptides

A common problem of recombinant expression is the production of misfolded proteins. Correct spatial organization and functionality of proteins require proper activity of chaperones/chaperonines and accurate disulfide bond formation, which is often difficult to achieve in a heterologous host. The flexible nature of the CF system permits the addition/subtraction of components, providing a controllable environment for protein folding, modification, and disulfide bond formation. The cytoplasm of *E. coli* contains numerous reductases, such as glutaredoxins and thioredoxins, and reducing agents—oxidized (GSSG) and reduced (GSH) glutathione in ratio 1:200—therefore, it is not suitable for the production of disulfide-bonded proteins [100]. On the other hand, bacterial periplasm contains a range of disulfide bond formation proteins (Dsb) that catalyze the creation and isomerization of cysteine bridges. For instance, DsbA induces disulfide creation in a consecutive manner [101], whereas DsbC isomerizes mis-oxidize, and, as a result, augments protein misfolding [102]. Similar to bacteria, eukaryotes contain protein disulfide isomerase (PDI) for disulfide bond formation and isomerization [103]. Therefore, disulfide bound formation in an *E. coli*-based CF system requires stabilization of the oxidizing environment and supplementation of the reaction mixture with disulfide isomerases, glutathiones, or different chaperones. The cell extract’s non-reducing conditions are achieved either by the application of iodoacetamide (IAM), which eliminates disulfide reducing activity and, consequently, allows folding of complex proteins, or by addition of GSSG and GSH at a ratio of 4:1 to stabilize a relatively oxidized sulfhydryl redox potential [104]. Moreover, modifying additives of spermidine and putrescine, which are known to influence the translational machinery, could increase the protein yield of the CF system via interaction with the ribosome, tRNAs, mRNA, and DNA [105]. The cell extract preparation procedure is also important for complex protein production in the cell-free system as the final lipid or, more precisely, vesicle concentration has been reported to influence oxidative phosphorylation, thereby providing an additional energy source for prolonged protein synthesis [106]. A stabilized non-reducing environment is not enough for proper disulfide formation. The abovementioned disulfide isomerases DsbC or PDI are required to shuffle incorrectly bonded disulfide bridges. The equilibrium between isomerases and the glutathione redox buffer helps maintain enzymes in the reduced state for successful cysteine framework formation [104,107]. Cytosolic chaperones play an important role in the general folding organization of proteins and could be added to the CF system to increase the yields of correctly folded compounds [108,109,110]. Chaperones DnaK/DnaJ (Hsp70s family) bind to nascent polypeptides on ribosomes and initiate protein folding, trafficking, and proteolytic degradation of misfolded molecules. Furthermore, chaperonins such as GroEL/GroES encapsulate non-aggregated misfolded proteins into its specialized cage containing a proper environment for folding. Both systems recognize exposed hydrophobic amino acid residues on the polypeptide surface and accelerate the folding rate over spontaneous folding [111]. The periplasmic chaperone Skp is also widely used in CF protein synthesis systems [106]. The *E. coli* Skp mediates outer membrane proteins (Omp) biogenesis and presentation of the mature polypeptide on the bacterial surface, as well as assists in folding of soluble proteins in the bacterial periplasm, preventing aggregation by acting as a holdase [112]. Summarizing the above, the synthesis of disulfide-bonded proteins in the cell-free system is a complex process that requires the creation of a special environment suitable for the oxidation of cysteines; high-yield polypeptide production depends on the addition of disulfide isomerases, creating proper cysteine bridges net and the usage of different adjuvant molecules (chaperones and chaperonins), thus optimizing amino acid chain spatial organization.

Notably, a CF system with enhanced folding activity can be successfully applied to a diverse range of proteins and peptides. Successful examples include the production of active enzymes with multiple disulfide bonds, such as plasminogen activator [104], urokinase protease [29], and various antibodies [107,108,113]. Another field of CF technology providing correct spatial organization is the production of complex proteins, such as virus-like particles (VLP) composing the outer shell of virus capsid [114,115]. Additionally, short antimicrobial peptides, such as cysteine-rich cationic beta-defensin-2 (hBD2) synthesized in a fusion form with thioredoxin [57], can be produced and correctly folded using the CF system.

### 3.4. Incorporation of NAAs into Proteins

Site-specific incorporation of NAAs with diverse physical, chemical, or biological properties into proteins provides scientists with a powerful tool for fundamental research and biotechnological application. The methodology allowing for co-translational NAAs incorporation (genetic code expansion) relies on the application of a non-endogenous translation system, referred to as orthogonal, OTS, and unique codon assigned for the NAAs [116]. OTS includes heterogenous aaRS, its cognate tRNA, and occasionally elongation factor Tu, EF-Tu [117]; this orthogonal set should act independently from the endogenous aa-RS/tRNA/codon set, but be compatible with translational apparatus of host cells [118]. One of the codons must be reassigned to encode NAA, and techniques based on the application of one of the stop codons (nonsense suppression), rarely used sense codons [119,120,121] (sense suppression), or four- or five-base extended codons (frameshift suppression) [122,123,124] have been developed. The most reliable and well-established methodology to genetically encode NAAs in *E. coli* is based on the application of amber stop codon, UAG, the most rarely used stop codon [125,126], recognized by the suppressor tRNA. The OTS is usually derived from tyrosyl-tRNA synthetase (TyrRS)/tRNATyr from *Methanocaldococcus jannaschii* [127], pyrrolysine-RS/tRNAPyl pair from *Methanosarcina species* [128,129,130], and others [118]. Recently the overall productivity of genetic code expansion in *E. coli* has been greatly improved due to tremendous progress in evolving more efficient orthogonal components and reducing or eliminating the negative effect of release factor 1, RF1, responsible for translation termination by binding to amber codon [131].

The CF system application for genetic code expansion has several advantages over in vivo processes. One of the problems of the genetic code expansion technique is the necessity to apply a time-consuming positive-negative selection procedure to evolve orthogonal aaRS that will specifically recognize and charge suppressor tRNA with the defined NAA [116,132,133]. The open nature of the CF system avoids this procedure by supplying the reaction medium with suppressor tRNA molecules chemically acylated with NAA [134]. It is also possible to supply the CF reaction with a diversity of preformed NAA-charged tRNA molecular species generated by artificial ribozymes, also called flexizymes, representing the highly flexible tRNA aminoacylation system [135,136].

A major limitation of genetic code expansion methodology in vivo is relatively low yield of recombinant protein with co-translationally incorporated NAA. The main reason for the low suppression efficiency is competition between the suppressor tRNA and RF1, which specifically recognizes UAA/UAG stop codons and terminates the translation process. Unfortunately, the simple removal of RF1 from the cell by inactivation of the *prfA* gene was not possible, since this gene is responsible for the viability of *E. coli* cells. One way to eliminate RF1’s deleterious effect on nonsense suppression efficiency is based on the selection of orthogonal ribosome/mRNA pairs. The Shine-Dalgarno and anti-Shine-Dalgarno sequence of orthogonal mRNA (o-mRNA) and 16S rRNA of the orthogonal ribosome (o-ribosome), respectively, are mutated, enabling strong pairing and independent functioning of the evolved components [137]. Since the A-site of 30S ribosome was mutated to provide a higher affinity to NAA-charged o-tRNA (ribo-X), suppression efficiency was facilitated by decreasing RF1 binding with the AUG codon [138]. Reasonably, these orthogonal ribosome/mRNA pairs can be applied to CF NAAs incorporation

The absence of a cell barrier in cell-free expression systems was successfully explored to directly sequestrate RF1 by binding with an RF1-specific RNA aptamer [139] or with anti-RF1 antibodies [140]. It is also possible to apply an RF1-withdrawn variant of the PURE system. Another way to prepare an RF1-depleted CF system relies on the removal of RF1, preliminarily tagged with C-terminal chitin-binding domains, from the reaction medium by the S30 lysate filtration through a chitin column [141]. The breakthrough in enhancing nonsense suppression efficiency both in vivo and in vitro was achieved by creating RF1-deficient *E. coli* strains with recoded genome: the TAG codons in the genomes of these strains were partially [142] or totally substituted [143] by TAA or TGA stop signals, allowing to knock out the *prfA* gene without negative effect on cell viability. The evolved RF1-depleted *E. coli* strains were applied in 30S lysate preparation for in vitro co-translational NAAs incorporation, obtaining RF1-depleted CF systems that were found to be highly productive and accurate [144,145,146].

The other reason for the relatively low nonsense suppression efficiency relies on the low affinity of orthogonal components to each other and whole OTS to the translational apparatus of host organisms. To achieve higher productivity of in vivo methodology, the efficiency of orthogonal components can be greatly improved either quantitatively by increasing gene copy number encoding OTS and, hence, elevating the overall concentration of orthogonal components inside the cell [147,148]; or qualitatively by selecting more efficient aaRS [149,150,151], evolving optimized suppressor tRNA with high affinity to endogenous EF-Tu [152], and by generating orthogonal EF-Tu [117,153]. The CF methodology enables to significantly increase the yield of NAA-incorporated protein by straightforward addition of an excess of orthogonal components to the reaction medium [154,155].

Although the application of a combination of evolved highly efficient OTS and RF1-depleted *E. coli* strain has been reported to enable multiple incorporations of identical NAAs into desired loci of target proteins [147,149], the incorporation of multiple diverse NAAs into the same protein is still challenging. The application of the CF methodology allows the incorporation of numerous NAAs with high productivity. Firstly, it is possible to assign two different stop codons for distinct NAAs; CF synthesis of such proteins was reported to occur without notable yield loss [156]. Secondly, the in vitro methodology enables the application of frame-shift suppression applying o-ribosome (ribo-Q) [123,129] or modified tRNA with 4-base anti-codon [157]. Utilization of sense codon suppression was successfully applied in vivo; however, the efficacy of this technique was demonstrated to depend on the competition of native and o-tRNA for sense codon binding [158]. The flexizyme application allows reassigning of multiple sense codons in vitro without significant loss in peptides or protein yields [135].

Finally, the cell-based methodology is not suitable for the incorporation of some NAAs, including those that cannot be transported inside the cell, can be metabolized, or are toxic. An application of the CF system is the single possibility to incorporate such NAAs [159,160].

To summarize, the CF methodology for incorporation of NAAs has significant advantages over in vivo processes: the productivity of the CF system is higher; the absence of barriers in the CF methodology enables enhancement of efficiency of NAAs incorporation by direct modification of the reaction medium; it is suitable for bulky, unstable, and toxic NAAs incorporation; and enables multiple incorporations of distinct NAAs.

### 3.5. Post-Translational Modifications

The fate of a protein in eukaryotic cell drastically depends on its post-translation modifications (PTM), such as acetylation, methylation, ubiquitination, phosphorylation, and glycosylation. The majority of CF synthesis of post-translationally modified recombinant proteins relies on eukaryotic cell extracts, while *E. coli* lysate application is limited due to the absence of PTM machinery types in this host organism [7,161]. Nonetheless, some progress has been achieved in glycosylation of recombinant proteins expressed in the *E. coli*-based CF system. Asparagine-linked (N-linked) glycosylation is the most widespread and common PTM [162], determining recombinant protein folding, stability, and catalytic activity. Protein glycosylation relies on the activity of oligosaccharyltransferase (OST) that transfers oligosaccharide from lipid-linked oligosaccharides (LLOs) to the asparagine in the conserved motif of the acceptor polypeptides [163]. The gene cluster *pgl*, protein glycosylation locus, of Gram-negative bacterium *Campylobacter jejuni* has been shown to encode the glycosylation system that could function efficiently in *E. coli* [164]. The addition of purified *C. jejuni* OST, *Cj*PglB, and LLO to the commercial in vitro protein synthesis system, either based on *E. coli* cell extract or PURE, resulted in the invention of glycoCFE and glycoPURE, respectively, enabling the production of glycosylated AcrA protein [165]. This approach was further modified by developing a glycoengineered *E. coli* strain, CLM24, expressing both *Cj*PglB OST and *Cj*LLOs that were used for cell extract preparation [166]. The resulting CF glycoprotein synthesis (CFGpS) system is cost-effective and easy to prepare since the glycosylation components do not require purification.

Although protein glycosylation in the *E. coli* CF system is the only successful example of PTM and considerable improvement is still required, there is hope for the expansion of controllable PTM. Other types of PTM were recently applied for recombinant protein modification in an *E. coli* background, including phosphorylation achieved due to the activity of expressed human Jun N-terminal kinase 1, JNK1 [167], sumoylation—in the presence of SUMO (small ubiquitin-like modifier) and SUMO E3 ligase [168], and methylation catalyzed by calmodulin lysine methyltransferase, CaM KMT [169]. Reasonably, in vitro application of the abovementioned PTM approach based on expression of modification machinery components is possible and has the potential to provide a reliable methodology for well-regulated protein modification.

## 4. Conclusions

Over the last years, CF protein expression methodology has become more productive, cost-effective, and applicable for large-scale production due to the invention of continuous-exchange technologies and utilization of less-expensive energy resources [41,42,48,170]. The technically advanced CF methodology for peptide/protein biomanufacturing has recaptured the attention of protein scientists. Highly flexible and reliable in vitro technologies have been successfully applied to the synthesis of difficult-to-obtain proteins, high-throughput protein library production, and directed protein evolution [7,171]. In vitro protein synthesis application is still limited because of the relatively high cost and necessity to adjust reaction medium components to achieve optimal conditions for target protein production. However, future investigation will help overcome these limitations and extend the potential of the CF synthesis methodology.

## Figures and Tables

**Figure 1 ijms-21-00928-f001:**
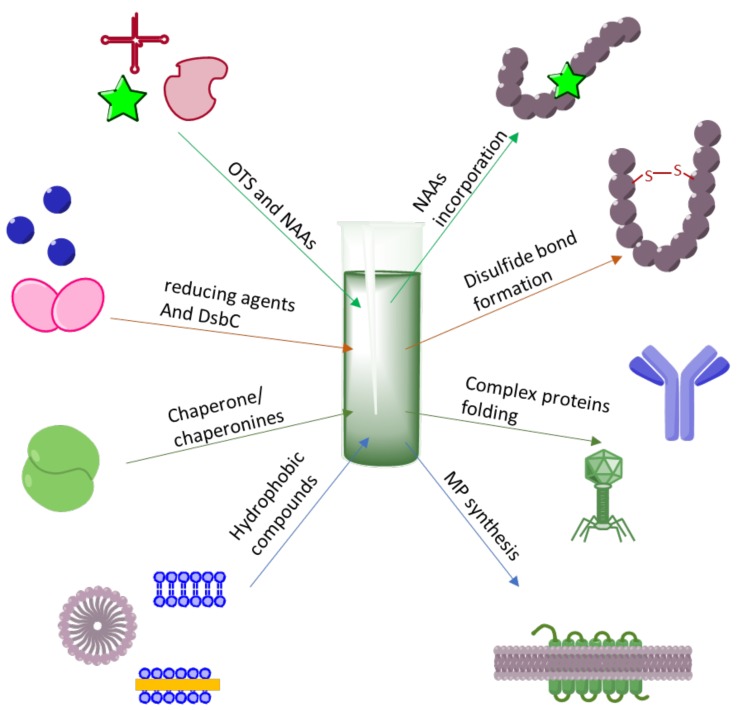
Schematic representation of CF methodology modification for the production of diverse proteins. Addition to the reaction mixture of orthogonal translation system (OTS) components and NAAs is required for the synthesis of site-specifically labeled proteins; reducing agents and disulfide-bond isomerase (DsbC) addition enables correct disulfide bond formation, while the presence of chaperons/chaperonins is required for complex proteins synthesis, such as antibody and virus-like particles (VLP); the presence of detergent micelles or lipid bicelles, membrane microsomes or nanodiscs is required for integral membrane protein (MP) synthesis.

**Figure 2 ijms-21-00928-f002:**
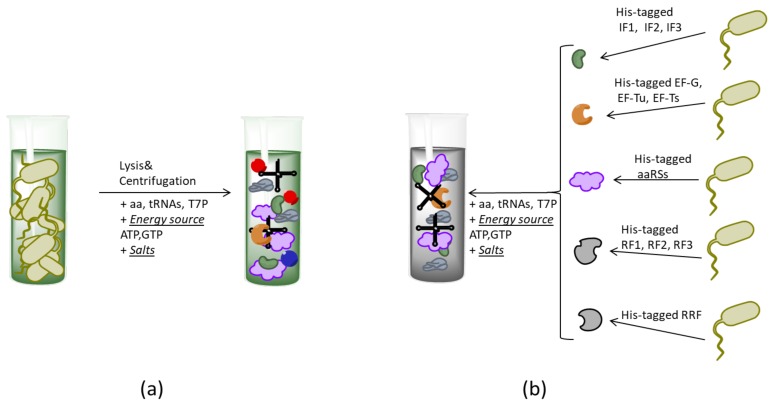
Schema of different types of CF system preparation. (**a**) The extract-based CF system is obtained from the clarified *E. coli* cells lysate supplemented by some other compounds and components of transcription/translation processes, including T7 polymerase (T7P), energy regeneration source, and salts. (**b**) The PURE CF methodology utilizes His-tagged *E. coli* enzymes transcription/translation factors of high purity (initiation IF, elongations EF, and release RF factors, ribosomal-regeneration factor (RRF), all type of aminoacyl-tRNA synthetases, aaRS, and other), preliminarily expressed and isolated from the heterologous host.

**Figure 3 ijms-21-00928-f003:**
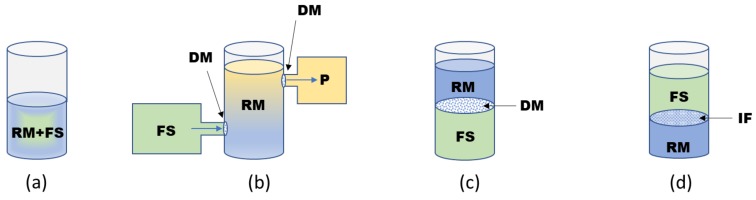
Schematic representation of the different formats of CF synthesis methodology. (**a**) The batch format utilizes one vesicle comprising both reaction medium (RM) and feeding solution (FS). (**b**) Continuous-flow CF methodology: FS is permanently pumped in RM through the dialysis membrane (DM); the newly synthesized proteins or by-products are removed from the RM by passing through the DM with other cut-off pores, allowing for target protein collection. (**c**) Continuous-exchange CF is based on dialysis through the membrane, allowing for the constant substrate addition and by-product removal from the RM. (**d**) Bilayer format: RM and FD represent two different liquid phases; the exchange and prolonged protein synthesis occur at the phase border or interphase (IF).

**Figure 4 ijms-21-00928-f004:**
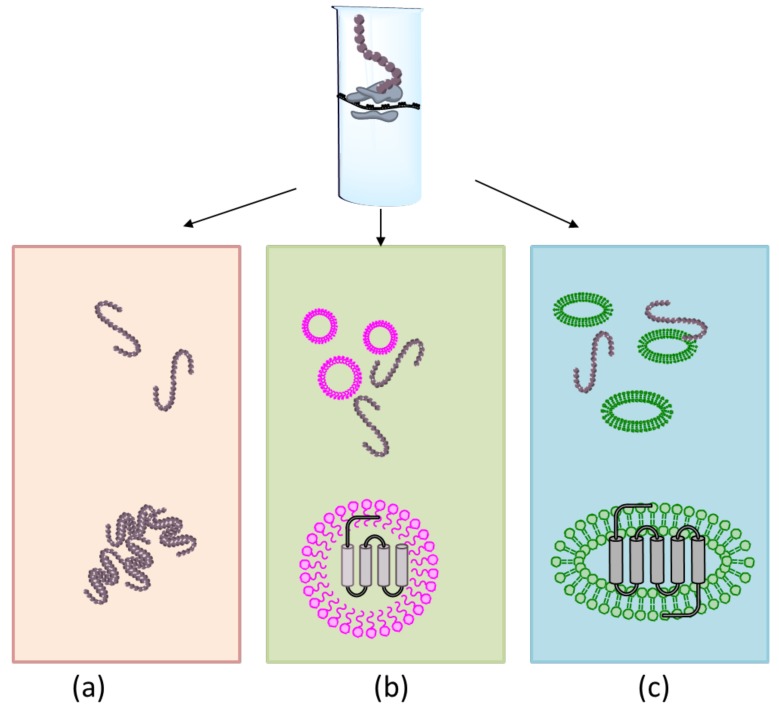
CF methodology for synthesis of MPs. (**a**) In the absence of hydrophobic compounds, the MPs can be produced as precipitates (P-CF). (**b**,**c**) The addition of detergents (D-CF) or lipids (L-CF) stimulates MPs synthesis in a soluble form.

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
