# Peer review of "Escherichia coli* Extract-Based Cell-Free Expression System as an Alternative for Difficult-to-Obtain Protein Biosynthesis"

_ijms, 2020, doi:10.3390/ijms21030928_

Round 1

Reviewer 1 Report

The review is very similar to reviews on the topic already published in 2019, and does not appear to add additional value to the field. Additionally, this review is lacking several key references on the topics it covers.

Author Response

Comments

The review is very similar to reviews on the topic already published in 2019, and does not appear to add additional value to the field. Additionally, this review is lacking several key references on the topics it covers. 

Response

We agree the cell-free protein expression methodology is widely explored by the scientific community, this results in a large number of both research and review publication appeared. We thoroughly analyzed recently published review and found out that most of them describe the cell-free methodology in term of its application to the synthetic biology and biomedical industry. The similar reviews describe very specific issues using highly professional vocabulary that makes them very difficult to understand for the broad variety of readers form the scientific community. We aimed to describe the cell-free methodology in more general terms and encourage scientist from the diverse field (including students studying protein engineering, medicine and biotechnology) to pay attention to this technology. We intended to focus on the description of the general principles of cell-free protein synthesis and methodological tools facilitating the production of the difficult-to obtain proteins. It is also worth to mention, that we provide a detailed historical overview providing a better understanding of cell-free methodology development. Concerning lacking references: we possibly missed some publication in such an intensively investigated field. So, we will be happy if the Reviewer 1 would point them up for the further improvement of our manuscript    

Reviewer 2 Report

The authors of the manuscript provide a nice and refreshing review of cell free synthesis. The two most recent reviews on the subject really focus of cell free synthesis in the light of synthetic biology, so it’s nice to see the topic updated in a more general context. I like that the authors reviewed the history of the subject (which hasn’t been updated in ~ 5 years), and how they broke down specific modifications to hard-to-express proteins in the latter section of the review. Overall, I think at the review is thorough and will add to the field. I have some comments to improve the readership/readability and scope that I believe will serve to improve the manuscript

Major comments:

I think it is important that the authors have the manuscript fully reviewed for proper sentence structure, grammar and punctuation. I think that this is important to increase readability. For example, in the abstract alone I noted the following. I am not sure if the lack of italics is due to a proofing/submission system limitation, but I am hoping that is can be addressed in a revised manuscript.

Abstract:

First sentence: I don’t understand what the authors mean by ‘an application.’ An application would suggest a use, but the rest of the sentence doesn’t really support the use of protein/peptides.

Second sentence: remove ‘the’. It should be just ‘recombinant protein expression….’ Escherichia should be italicized in this sentence as well. Remove ‘s’ from organisms (E. coli is only one genus/species)

Third sentence: “as well as due to the requirement for the special modification for proper folding and activity.”  I think that this should be its own sentence and should read something along the lines of “In addition, many proteins and peptides require special modification for proper folding and activity.”

Fourth sentence: in vitro should be italicized

Six: remove prokaryotic because it is evident that E. coli is a prokaryote. In vitro needs to be italicized.   “and modification of in vitro approach required for the difficult-to-obtain protein production.” This sentence should read ‘and the modification of in vitro approaches required for difficult to obtain protein production’

These are just a few of the errors that I found throughout the manuscript. Please have a few colleagues read it over to make sure your well scoped and thoroughly researched paper is not lost on spelling, grammar and punctuations issues.

In the introduction, the authors do a great job at highlighting the benefits of cell free systems. However, their last point (Finally, the in vitro approach is a salvation for the synthesis of toxic peptides or proteins) is not really expanded upon. A naïve reader may not immediately understand why this is the case. Can the authors elaborate briefly on this point as they did for points 1 and 2 in the paragraph?

As the authors acknowledge, side reactions and the accumulation of undesirable molecules can reduce efficiency. However, the authors only touched upon this lightly. I think for a well rounded review, the authors should at least acknowledge recent developments in CF synthesis where compartmentalization occurs in liposomes/nanoparticles (https://pubs.rsc.org/en/content/articlelanding/2016/cc/c6cc00223d/unauth#!divAbstract) and microfluidic devices (https://onlinelibrary.wiley.com/doi/abs/10.1002/cbic.200400321).

I think it would be important to defined what an orthogonal translation systems (OTS) prior to figure 1. It is an important concept, and it should be explained in general terms so that individuals outside the field can understand the concept. A figure demonstrating the concept might also help.

The history section does a great job summarizing the flow of accomplishments. However, I think that this approach is also worth mentioning because it appears to have eased the preparation of CF extracts significantly “https://www.ncbi.nlm.nih.gov/pubmed/29131146)

In the figure legend for fig 1, what does MP stand for? VLP is also defined but not used in the image (?)

In general, in vitro should be italicized throughout.

Author Response

Response to Reviewer 2

We would like to thank the reviewer 2 for the time and valuable comments. Further, we provide point-by-point response to all of the issues raised in the reviews comments is below.

Point 1: Major comments:

I think it is important that the authors have the manuscript fully reviewed for proper sentence structure, grammar and punctuation. I think that this is important to increase readability. For example, in the abstract alone I noted the following. I am not sure if the lack of italics is due to a proofing/submission system limitation, but I am hoping that is can be addressed in a revised manuscript.

Response 1: The manuscript was thoroughtly reviewed and the English editing of the manuscript was performed.

Point 2: Abstract:

First sentence: I don’t understand what the authors mean by ‘an application.’ An application would suggest a use, but the rest of the sentence doesn’t really support the use of protein/peptides.

Response 2: The abstract was rewritten and corrected for the better understanding

Point 3: Second sentence: remove ‘the’. It should be just ‘recombinant protein expression….’ Escherichia should be italicized in this sentence as well. Remove ‘s’ from organisms (E. coli is only one genus/species)

Response 3: Corrected accordingly

Point 4: Third sentence: “as well as due to the requirement for the special modification for proper folding and activity.”  I think that this should be its own sentence and should read something along the lines of “In addition, many proteins and peptides require special modification for proper folding and activity.”

Response 4: Corrected accordingly

Point 5: Fourth sentence: in vitro should be italicized

Response 5: Corrected.

Point 6: Six: remove prokaryotic because it is evident that E. coli is a prokaryote. In vitro needs to be italicized.   “and modification of in vitro approach required for the difficult-to-obtain protein production.” This sentence should read ‘and the modification of in vitro approaches required for difficult to obtain protein production’

Response 6: Corrected.

Point 7: These are just a few of the errors that I found throughout the manuscript. Please have a few colleagues read it over to make sure your well scoped and thoroughly researched paper is not lost on spelling, grammar and punctuations issues.

Response 7: The English editing of the manuscript was performed.

Point 8: In the introduction, the authors do a great job at highlighting the benefits of cell free systems. However, their last point (Finally, the in vitro approach is a salvation for the synthesis of toxic peptides or proteins) is not really expanded upon. A naïve reader may not immediately understand why this is the case. Can the authors elaborate briefly on this point as they did for points 1 and 2 in the paragraph?

Response 8: We added a brief explanation of how the heterologouse expression of moderately and highly toxic peptides/proteins lead to the cellular metabolism attenuation or death, respectively.

Point 9: As the authors acknowledge, side reactions and the accumulation of undesirable molecules can reduce efficiency. However, the authors only touched upon this lightly. I think for a well rounded review, the authors should at least acknowledge recent developments in CF synthesis where compartmentalization occurs in liposomes/nanoparticles (https://pubs.rsc.org/en/content/articlelanding/2016/cc/c6cc00223d/unauth#!divAbstract) and microfluidic devices (https://onlinelibrary.wiley.com/doi/abs/10.1002/cbic.200400321).

Response 9: The application of microfluidic devices for the enhanced productivity of the cell-free expression system and for the creaction of encapsulated (compartmentalized) artificial cells is briefly described (subsection 2.2, lines 214-223).

Point 10: I think it would be important to defined what an orthogonal translation systems (OTS) prior to figure 1. It is an important concept, and it should be explained in general terms so that individuals outside the field can understand the concept. A figure demonstrating the concept might also help.

Response 10: We added a short description of OTS (lines 64-65). The more detailed explanation of the genetic code expansion methodology exploring OTS and translational apparatus of the host cell is given in subsection 3.4. We hope that the readers outsite this field of research can find more detailed information (including figures) in the papers cited in the given manuscript. We also provided the readers by reference to our review article (ref [131]) describing NAAs incorporation methodology and recent advances in this area. 

Point 11: The history section does a great job summarizing the flow of accomplishments. However, I think that this approach is also worth mentioning because it appears to have eased the preparation of CF extracts significantly “https://www.ncbi.nlm.nih.gov/pubmed/29131146).

Response 11: The approach simplifiying the PURE system preparation is described (line 140-154).

Point 12: In the figure legend for fig 1, what does MP stand for? VLP is also defined but not used in the image (?)

Response 12: Corrected accordingly. The image of a virus in figure 1 depicts a virus-like particle (VLP).

Point 13: In general, in vitro should be italicized throughout

Response 13: Corrected.

Round 2

Reviewer 1 Report

It is reasonable to suggest that one might have overlooked a reference or two in the process of writing, but this reviewer wonders whether the authors are truly informed of the field given the magnitude of the missing references. Only 5 of 171 references are from 2019, and therefore the manuscript does not include the most recent advances published in 2018-2019. For example, none of the articles from the Methods & Protocols special issue on cell-free are cited. This is a problem since several of those publications discuss the production of difficult to make proteins and non-standard amino acid incorporations.

 At a minimum, the authors should read and cite the following:
https://doi.org/10.3390/mps2010024

https://doi.org/10.3390/mps2020052

https://doi.org/10.3390/mps2020028

https://doi.org/10.3390/mps2010016

Additionally, the authors should read the following 2 reviews from 2019. They will find that a significant fraction of their manuscript is redundant with these very recent reviews. In fact, some figures in the manuscript have uncanny resemblance to the figures in these publications.

http://dx.doi.org/10.1007/s00253-019-09690-6

https://doi.org/10.3390/mps2010024

In light of the gaps, and lack of up-to-date information in the review, it is not obvious that the manuscript adds much value. Additionally, the redundancy of the content with recent reviews reduces the utility of the review. If the authors feel that this manuscript does add value, they should identify the content that is distinct from recent reviews and reframe their manuscript around that content.